# Retinal Microcirculation and Cytokines as Predictors for Recurrence of Macular Edema after Intravitreal Ranibizumab Injection in Branch Retinal Vein Occlusion

**DOI:** 10.3390/jcm10010058

**Published:** 2020-12-26

**Authors:** Hidetaka Noma, Kanako Yasuda, Tatsuya Mimura, Noboru Suganuma, Masahiko Shimura

**Affiliations:** 1Hachioji Medical Center, Department of Ophthalmology, Tokyo Medical University, Tokyo 193-0998, Japan; kana6723@yahoo.co.jp (K.Y.); h-ort@tokyo-med.ac.jp (N.S.); masahiko@v101.vaio.ne.jp (M.S.); 2Department of Ophthalmology, Teikyo University School of Medicine, Tokyo 173-8606, Japan; mimurat-tky@umin.ac.jp

**Keywords:** relative flow volume, laser speckle flowgraphy, macular edema, branch retinal vein occlusion, monocyte chemotactic protein-1, interleukin-8, interferon-inducible 10-kDa protein

## Abstract

Purpose: To investigate the relationship between retinal blood flow, presence or absence of recurrence of macular edema, and levels of cytokines, after intravitreal ranibizumab injection (IRI) in patients with branch retinal vein occlusion (BRVO). Methods: In 47 patients with BRVO and macular edema, we used laser speckle flowgraphy (LSFG) to measure the relative flow volume (RFV) of the retinal arteries and veins passing through the optic disc in the occluded and non-occluded regions of the retina before and after IRI. Aqueous humor samples were obtained at the time of IRI. Levels of vascular endothelial growth factor (VEGF), soluble VEGF receptor (sVEGFR)-1, sVEGFR-2, placental growth factor (PlGF), platelet-derived growth factor (PDGF)-AA, soluble intercellular adhesion molecule (sICAM)-1, monocyte chemoattractant protein 1 (MCP-1), interleukin (IL)-6, IL-8, IL-12 (p70), IL-13 and interferon-inducible 10-kDa protein (IP-10) were measured by the suspension array method. Patients were categorized into two groups on the basis of whether or not macular edema recurred at 2 months after IRI: the nonrecurrent group, *n* = 24; and the recurrent group, *n* = 23. Results: In the veins of the occluded region, RFV showed a significant difference between baseline and 1 month after IRI (*p* < 0.001) in the recurrent group and the percent change of RFV showed a significant difference between the recurrent and nonrecurrent groups (*p* = 0.005). Furthermore, we found a significant negative correlation between RFV in the veins of the occluded region and aqueous levels of MCP-1, IL-8 and IP-10 at baseline (*p* = 0.029, *p* = 0.035, and *p* = 0.039, respectively). In the recurrent group, the arteries and veins of the non-occluded and occluded regions showed no significant association between RFV and the aqueous levels of any factors. Conclusions: These findings suggested that a decrease in RFV in the veins of the occluded region might be associated with the recurrence of macular edema and that the recurrence might depend on the change in RFV in the veins of the occluded region rather than the levels of cytokines.

## 1. Introduction

Branch retinal vein occlusion (BRVO) is a common retinal vascular condition in patients with lifestyle-related diseases such as hypertension and arteriosclerosis. Because retinal arterioles and venules have a common adventitia at arteriovenous crossings, the venule walls can be compressed by arteriosclerotic changes. Luminal narrowing leads to disturbance of laminar blood flow, and endothelial damage is caused by shear stress; subsequent thrombus formation leads to BRVO [1]. In the acute phase of BRVO, pressure increases in the capillaries and venules affected by the obstruction, leading to breakdown of the blood–retinal barrier and leakage of blood components. The main cause of visual impairment in patients with BRVO is macular edema. In macular edema, fluid accumulates in the inner to outer plexiform layer in the retina, which may be due to an alteration of retinal blood flow [2]. Thus, understanding the retinal hemodynamic abnormalities that underlie the pathogenesis of BRVO is of critical importance.

Laser speckle flowgraphy (LSFG) is a noninvasive, real-time method that can measure the relative blood flow on the optic nerve head [3,4]. The relative flow volume (RFV) obtained by LSFG was reported to be an accurate and reliable index of the blood flow volume in the retinal vessels [5,6]. Fukami et al. recently reported that RFV changed after intravitreal ranibizumab injection (IRI) [7]. Anti-VEGF (vascular endothelial growth factor) therapy such as IRI has become the first choice for macular edema in patients with BRVO [8,9,10]. However, recurrence of macular edema is frequent after one dose of IRI. We previously reported that the aqueous humor levels of several inflammatory factors are related to the severity of macular edema [11] and that the number of IRIs was significantly correlated with the baseline aqueous levels of several inflammatory factors [12], suggesting that inflammation may be involved in the recurrence of macular edema. However, it is unclear whether retinal blood flow is involved in the recurrence of macular edema and whether inflammation and inflammatory factors are involved in retinal blood flow. Accordingly, we evaluated retinal blood flow before and after IRI in patients with BRVO and examined the relationship between retinal blood flow and the presence or absence of recurrence of macular edema and levels of cytokines. To do so, we measured the RFV of the retinal arteries and veins by LSFG in the occluded and non-occluded regions, and investigated relations between the alterations of these flows and the presence or absence of recurrence of macular edema after IRI, in patients with BRVO. In addition, we assessed whether changes in these flows affected aqueous levels of cytokines at the time of IRI.

## 2. Methods

### 2.1. Patients

Patients were recruited at the Department of Ophthalmology, Tokyo Medical University. The study was approved by the University Ethics Committee (IRB No. H-132) and was performed in accordance with the Declaration of Helsinki. All participants provided written informed consent to participate in the study. This study was registered in the University hospital Medical Information Network (UMIN) clinical trials registry (UMIN000030311). All consecutive patients with BRVO who presented to Tokyo Medical University between June 2017 and July 2019 were screened using the criteria listed below. A final 47 patients with BRVO (47 eyes), who were scheduled to undergo IRI (Lucentis; 0.5 mg in 0.05 mL; Genentech, Inc., South San Francisco, CA, USA), were included. Criteria for receiving IRI were macular edema involving the fovea (central macular thickness (CMT) > 300 μm) and a best-corrected visual acuity (BCVA) less than 25/30. Exclusion criteria were BRVO of a macular venule, glaucoma, aphakia, rubeosis iridis, diabetes mellitus with diabetic retinopathy, clinically significant cataract, ocular infection, previous ocular inflammation, vitreous hemorrhage, a history of other retinal diseases, a refractive error below −6.0 diopter, a history of macular laser photocoagulation, administration of anti-inflammatory comedication, and administration of anti-VEGF agents by intravitreal injection. All patients were followed for at least 2 months after IRI. The patients were categorized into 2 groups on the basis of whether or not macular edema recurred at 2 months after IRI (nonrecurrent group, *n* = 24; and recurrent group, *n* = 23). Recurrent macular edema was defined as an increase in the CMT of more than 100 μm and a loss of visual acuity at 2 months after IRI. At the time of IRI, a mean volume of 0.1 mL of aqueous humor was collected by anterior chamber limbal paracentesis with a 30-gauge needle attached to an insulin syringe.

### 2.2. Routine Evaluations

A full ophthalmic evaluation was performed once a month for the duration of the study. These monthly evaluations included decimal BCVA, fluorescein angiography (FA; Digital Retinal Camera CF-1; Canon, NY, USA), and spectral-domain OCT (Spectralis, Heidelberg Engineering, Heidelberg, Germany). At the end of the study, patients underwent a follow-up evaluation consisting of decimal BCVA and spectral-domain OCT. Computer software was used to automatically assess the CMT, i.e., the distance from the inner limiting membrane to the retinal pigment epithelium, including any serous retinal detachments. For the statistical analysis, we converted the decimal BCVA to the logarithm of the minimum angle of resolution (logMAR).

### 2.3. Laser Speckle Flowgraphy (LSFG)

LSFG (Softcare, Fukutsu, Japan) was performed according to the previously published method [4]. This assessment is based on the speckled pattern that results from the reflection of laser light by the ocular fundus and the blurring of the pattern by the erythrocytes in the vessels of the fundus. The speckle contrast pattern can be visualized with a fundus camera with a diode laser (830-nm wavelength) and a charge-coupled device sensor (750 × 360 pixels). Furthermore, the variation in blurring can be used to calculate the mean blur rate (MBR), an index of relative blood flow velocity (expressed in arbitrary units, AU) [5,6].

In this study, we acquired 30 frames per second for 4 s and created a composite map of blood flow by calculating the mean flow. We calculated RFV by manually choosing one region of interest with a retinal blood vessel at its center (Figure 1) [13].

Then, we calculated a separate MBR for this vessel as the difference between the background choroidal blood flow and the overall MBR. Next, we used the LSFG analyzer software (version 3.1.6, Fukuoka, Japan) to determine the RFV by inputting the MBR of vessels that were transverse to the chosen retinal vessel (Cross-Section Ex). We calculated the RFV of the artery and vein passing through the optic disc in the occluded and non-occluded regions. Assessments were performed before IRI and 1 and 2 months afterwards. The RFV is hypothesized to accurately reflect blood flow in the top layer of the retina. In addition, we confirmed that the positioning for measurements at 1 and 2 months was the same as at baseline in all patients. Nonetheless, RFV measurements have been reported to have high reproducibility regarding measurement error [5,14].

The pupil was dilated with 0.5% tropicamide and 0.5% phenylephrine hydrochloride 20 min before LSFG. To evaluate changes of retinal blood flow, the percent change of RFV (% RFV) was calculated as follows: % RFV = (RFV_1month_ − RFV_baseline_)/RFV_baseline_ × 100, where RFV_baseline_ and RFV_1 month_ were the levels of RFV corresponding to the RFV at baseline and at 1 month after IRI, respectively.

### 2.4. Hemodynamics

In healthy individuals with normal eyes, RFV shows a bilinear relationship to ocular perfusion pressure (OPP) over a particular range [15]. We calculated OPP by assessing mean blood pressure (MBP) and measuring intraocular pressure (IOP) to ensure that physiological responses were excluded. MBP was based on systolic blood pressure (SBP) and diastolic blood pressure (DBP) and calculated with the following equation: MBP = DBP + 1/3(SBP − DBP). As a final step, we calculated OPP as follows: OPP = 2/3MBP − IOP.

### 2.5. Assessment of Cytokines and Growth Factors

To detect soluble VEGF receptor (sVEGFR)-1, sVEGFR-2, VEGF, placental growth factor (PlGF), platelet-derived growth factor (PDGF)-AA, soluble intercellular adhesion molecule (sICAM)-1, monocyte chemoattractant protein 1 (MCP-1), interleukin (IL)-6, IL-8, IL-12 (p70), IL-13, and interferon-inducible 10-kDa protein (IP-10), we used capture bead kits (Beadlyte; Upstate Biotechnology, Lake Placid, NY, USA), according to the manufacturer’s instructions. For measurements of PlGF and sICAM 1, we incubated undiluted aqueous humor (25 μL) for 16 to 18 h overnight, and for measurements of the remaining factors we incubated undiluted aqueous humor samples for 2 h; we performed all incubation steps in a dark room at room temperature. For each cytokine, we created duplicate standard curves from the reference set of concentrations in each kit. We analyzed samples with suspension array technology (xMAP; Luminex Corp, Austin, TX, USA) [11,16]. For each participant, to increase the accuracy of the measurements we assessed the samples from before and after the intervention at the same time and included control samples. The levels of the assessed factors in the aqueous humor samples were all above the following minimum detectable concentrations of the assay: sVEGFR-1, 1.59 pg/mL; sVEGFR-2, 44.81 pg/mL; VEGF, 0.64 pg/mL; PlGF, 0.37 pg/mL; sICAM-1, 0.03 ng/mL; MCP-1, 1.2 pg/mL; PDGF-AA, 0.64 pg/mL; IL-6, 0.29 pg/mL; IL-8, 0.14 pg/mL; IL-12 (p70), 0.14 pg/mL; IL-13, 0.12 pg/mL; and IP-10, 0.55 pg/mL.

### 2.6. Statistical Analysis

For all analyses, we used SAS System 9.4 software (SAS Institute Inc., Cary, NC, USA). Results are presented as means and SD. We used an unpaired Student’s *t* test to compare unpaired continuous variables and a paired Student’s *t* test to compare paired continuous variables (baseline vs. 1 month after IRI and 1 month vs. 2 months after IRI). We tested for correlations between variables with Pearson’s correlation analysis. A significant difference was assumed if two-tailed *p* values were less than 0.05.

## 3. Results

The patients with BRVO included 27 men and 20 women aged 65.2 ± 9.8 years (mean ± SD). The mean duration of macular edema was 42.5 ± 23.3 days (range, 9–20 days). Thirty-four of the 47 patients (72%) had hypertension, which was defined as current treatment with antihypertensive drugs or a blood pressure greater than 140/90 mmHg, and 22 of the 47 patients (47%) had hyperlipidemia. The mean SBP, DBP, MBP and OPP were 142 ± 16 mmHg, 85 ± 12 mmHg, 104 ± 11 mmHg, and 56 ± 7.8 mmHg, respectively. The mean baseline BCVA was logMAR 0.45 ± 0.37, and the mean baseline CMT was 638 ± 200 μm. We found no significant differences in the baseline values of any clinical parameters (age, sex, duration of macular edema, hypertension, SBP, DBP, hyperlipidemia, BCVA, CMT, MAP and OPP) between the nonrecurrent and recurrent groups (Table 1).

After IRI, BCVA improved significantly over time (1 month, 0.17 ± 0.32; 2 months, 0.18 ± 0.33; *p* < 0.001) and CMT decreased significantly over time (1 month, 240 ± 51 μm; 2 months: 405 ± 177 μm; *p* < 0.001).

At baseline, the arteries and veins of the non-occluded and occluded regions showed no significant correlation between RFV and the aqueous levels of any factors (Table 2). The veins of the occluded region showed a significant negative correlation between RFV and the aqueous levels of MCP-1, IL-8 and IP-10 (Table 2). In contrast, we found no correlation between RFV and the aqueous levels of sVEGFR-1, sVEGFR-2, VEGF, PlGF, sICAM-1, PDGF-AA, IL-6, IL-12, or IL-13 (Table 2).

In the whole sample, the arteries of the non-occluded and occluded regions showed no significant differences in RFV between baseline and 1 month after IRI (Figure 2A). The veins of the non-occluded region also showed no significant difference in RFV between baseline and 1 month after IRI, but the veins of the occluded region did (Figure 2A). In the nonrecurrent group, the arteries and veins of the non-occluded and occluded regions showed no significant differences in RFV between baseline and 1 month after IRI (Figure 2B). In the recurrent group, the arteries of the non-occluded and occluded regions and the veins of the non-occluded region also showed no significant differences in RFV between baseline and 1 month after IRI, but the veins of the occluded region did show a significant difference (Figure 2C).

In the whole sample, the arteries and veins of the non-occluded and occluded regions showed no significant differences in RFV between 1 and 2 months after IRI (Figure 3A). This was also the case in the nonrecurrent group (Figure 3B) and the recurrent group (where 1 month was before the first recurrence and 2 months was at recurrence; Figure 3C).

The arteries of the non-occluded and occluded regions showed no significant differences in the percent change of RFV between the nonrecurrent and recurrent groups (Figure 4). The veins of the non-occluded region also showed no significant difference in the percent change of RFV between the nonrecurrent and recurrent groups, but the veins of the occluded region did (Figure 4).

At recurrence, RFV was not significantly associated with the aqueous levels of any factors in the arteries or veins of the non-occluded and occluded regions (Table 3).

## 4. Discussion

This study found a significant difference in RFV between baseline and 1 month after IRI in the veins of the occluded region, suggesting that ranibizumab decreases retinal blood flow in these vessels. This finding is in agreement with the study by Fukami et al. [7], who found that ranibizumab induces retinal vessel vasoconstriction, as did other studies [7,17,18]. This vasoconstriction could further increase vascular resistance in the occluded region, which might decrease retinal blood flow in the veins of that region. Interestingly, in our study the recurrent group, but not the nonrecurrent group, showed a significant decrease in RFV in the veins of the occluded region 1 month after IRI compared with baseline. These results suggest that a decrease in retinal blood flow in the veins of the occluded region might influence the recurrence of macular edema. Previously, development of macular edema in patients with BRVO was hypothesized to be associated with fluid flux from the vascular compartment to the tissues (Starling’s Law) after breakdown of the blood–retinal barrier [19,20]. This hypothesis is supported by the report that a decrease in retinal blood flow in the veins of the occluded region may lead to stagnation of retinal blood flow, resulting in recurrence of macular edema [21]. Taken together with the present findings, such reports confirm that macular edema might be due to stagnation of blood flow in the veins of the occluded region in patients showing recurrence of macular edema after IRI. In fact, we found a significant difference in the percent change of RFV between the nonrecurrent and recurrent groups in the veins of the occluded region, suggesting that a decrease in retinal blood flow in these veins might be related to the recurrence of macular edema. Therefore, a change of retinal blood flow in the veins of the occluded region, as measured by LSFG, could be an indicator of recurrence of macular edema. As a new strategy, measuring RFV in the veins of the occluded region by LSFG may help predict the recurrence of macular edema. In clinical practice, if RFV decreases, as measured with LSFG, taking circulatory drugs might help prevent a decrease in RFV. However, further investigation will be required to elucidate the relation between taking circulatory drugs, change in RFV, and recurrence of macular edema.

We also found a significant negative correlation between RFV in the veins of the occluded region and the aqueous levels of MCP-1, IL-8 and IP-10 in patients with BRVO and macular edema, suggesting that these factors can influence blood flow in the retinal veins of the occluded region in these patients. MCP-1 is a chemokine that recruits monocytes/macrophages into tissues [22]. Moreover, the recruitment of monocytes/macrophages to vessel walls promotes vascular permeability, potentiating macular edema [23,24]. IL-8 is a potent chemoattractant that activates neutrophils and T cells. Production of IL-8 is induced by exposure of vascular endothelial cells to hypoxia and oxidative stress [25,26,27], and IL-8 was reported to promote the adhesion of leukocytes to the vascular endothelium [28,29]. We previously reported that the aqueous humor levels of MCP-1 and IL-8 were significantly correlated with each other in patients with BRVO [11]. IP-10 has an antiangiogenic effect by inhibiting the proliferation of ECs and induction of EC apoptosis [30]. It is also a CXC chemokine that is secreted by macrophages, endothelial cells, and fibroblasts, acts as a chemoattractant for macrophages, dendritic cells, and T-cells, contributes to T-helper type 1 immune responses and activates cell-mediated immunity in general [30,31]. The level of IP-10 was reported to be significantly higher in patients with central retinal vein occlusion [32]. After retinal vein occlusion in vivo, leukocytes show increased rolling and adhesion to vein walls, which leads to stagnant blood flow [33]. Thus, the entrapment of leukocytes associated with increased rolling and adhesion of these cells by these inflammatory factors may reduce RFV in patients with BRVO and macular edema.

Previously, we reported that inflammatory factors were significantly decreased after IRI [34,35], so, in the present study, we expected that the RFV would be increased after IRI; however, this was not the case. In the present study, only 3 inflammatory factors (MCP-1, IL-8 and IP-10) were significantly negatively correlated with RFV. Ranibizumab induces retinal vessel vasoconstriction [7,17,18], and vascular endothelial growth factor can cause an increase in retinal blood flow [36], probably via the production of nitric oxide [37]. Nitric oxide plays an important role in the vasodilation of retinal vessels [38] and the increase in the retinal blood flow and velocity. Therefore, in contrast to VEGF, ranibizumab is more likely to induce vasoconstriction of vessels and a decrease in retinal circulation. Thus, rather than a change in inflammatory factors, a change in RFV might be related to vasoconstriction after IRI.

On the other hand, in the arteries of the non-occluded and occluded regions, we found no significant differences of RFV between baseline and 1 month after IRI in either the nonrecurrent or the recurrent group. Our findings in the retinal arteries may be explained by autoregulation of the tone of retinal arterioles. Retinal blood flow is regulated by vasoactive substances released by the vascular endothelium and retinal tissues that surround the arterioles, i.e., through myogenic and metabolic mechanisms [39]. These vasoactive substances change the tone of the arterioles and capillaries in response to changes in OPP or metabolic requirements of the tissue. For example, if arterioles are mechanically stretched or arteriolar transmural pressure increases then contractile factors are released that change the tone of arteriolar smooth muscle cells and capillary pericytes [39]. An earlier study provided evidence that retinal arterioles are a key factor in autoregulating retinal blood flow through a myogenic mechanism [40].

This study found no significant differences of RFV between 1 and 2 months after IRI in the arteries and veins of the non-occluded and occluded regions in either the nonrecurrent or recurrent group. This finding suggests that a change in retinal blood flow before and after recurrence may not actually affect the recurrence of macular edema. Furthermore, at recurrence we found no significant association between RFV and the aqueous levels of any factors in the arteries and veins of the non-occluded and occluded regions, suggesting that levels of cytokines at recurrence of macular edema might not influence retinal blood flow. Taken together, the above results and ours indicate that the recurrence of macular edema might depend on the change in retinal blood flow in the veins of the occluded region from baseline to 1 month after IRI rather than the change in retinal blood flow before and after recurrence of macula edema or the change in the levels of cytokines. However, a larger, prospective study will be required to confirm the influence of cytokines on RFV and recurrence of macular edema.

The present study had the following limitations. First, our study population was relatively small. Second, we were not able to collect aqueous humor 2 months after IRI in the nonrecurrent group. Therefore, we were unable to investigate the difference in cytokines between the nonrecurrent group and the recurrent group. Further investigation with a larger sample size is required to clarify the relationship of cytokines with RFV and recurrence of macular edema.

In conclusion, we found that the group with recurrent macula edema showed a significant difference in RFV between baseline and 1 month after IRI in the veins of the occluded region and that the percent change of RFV in the veins of the occluded region was significantly different between the nonrecurrent and recurrent groups. Furthermore, we found a significant negative correlation between RFV and the aqueous levels of MCP-1, IL-8 and IP-10 at baseline but no correlation between RFV and any cytokine levels at recurrence. These findings suggest that a decrease in retinal blood flow in the veins of the occluded region might be associated with the recurrence of macular edema and that the recurrence might depend on the change in retinal blood flow in the veins of the occluded region rather than the levels of cytokines.

## Figures and Tables

**Figure 1 jcm-10-00058-f001:**
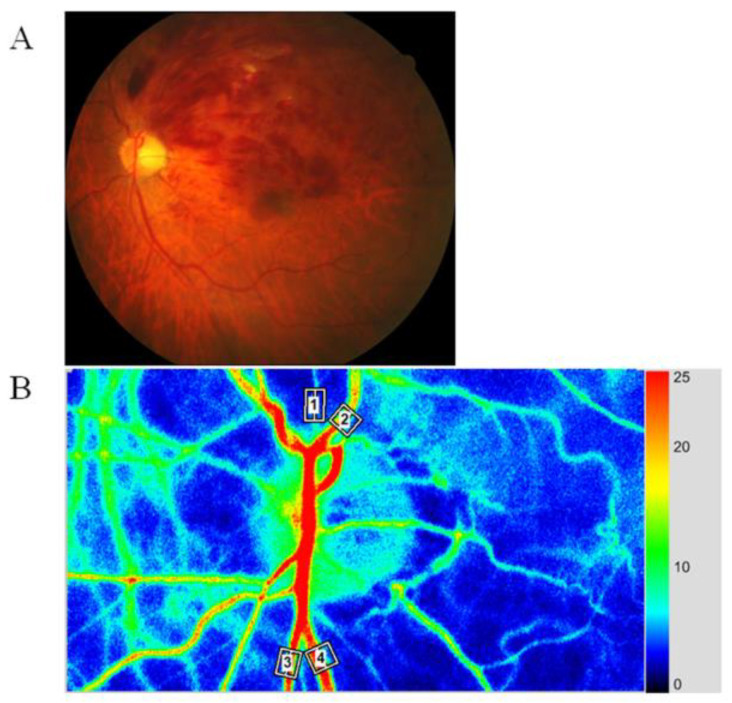
Representative fundus color photograph, and representative relative blood volume data obtained with laser speckle flowgraphy (LSFG). (**A**) Fundus color photograph shows branch retinal vein occlusion (BRVO). The upper part is the occluded region, and the lower part is the non-occluded region. (**B**) Blood flow was automatically tracked in an artery (white square #1) and vein (white square #2) in the occluded region and in an artery (white square #3) and vein (white square #4) in the non-occluded region.

**Figure 2 jcm-10-00058-f002:**
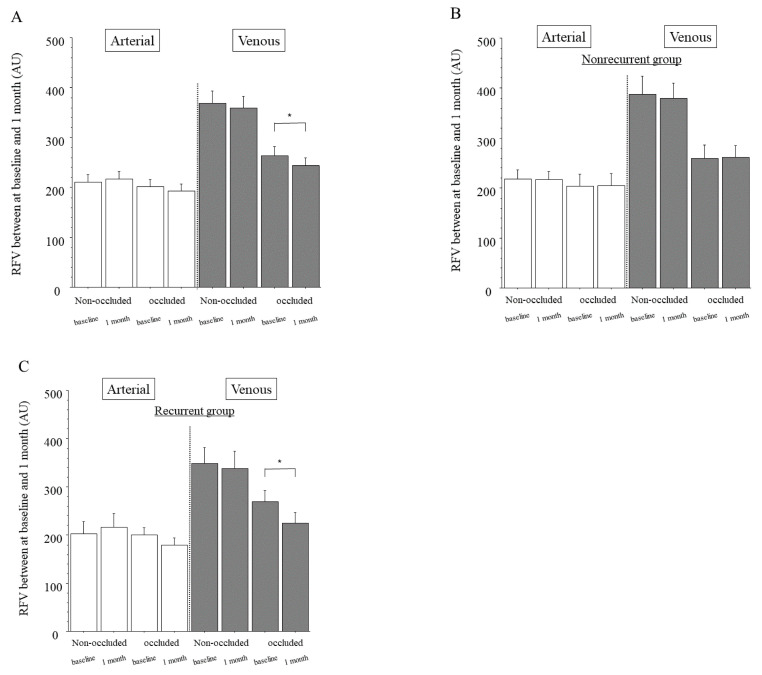
Relative blood volume between baseline and 1 month after intravitreal ranibizumab injection. (**A**) In the whole sample, we found no significant differences of relative flow volume (RFV) between baseline and 1 month after intravitreal ranibizumab injection (IRI) in the arteries of the non-occluded (*p* = 0.593) or occluded regions (*p* = 0.245). The veins of the non-occluded region showed significant differences in RFV between baseline and 1 month after IRI (*p* = 0.501), but the veins of the occluded region did not (* *p* = 0.029). (**B**) In the group with no recurrence of macula edema (nonrecurrent group, *n* = 24), we found no significant differences of RFV between baseline and 1 month after IRI in the arteries and veins of the non-occluded and occluded regions (arteries: *p* = 0.918, *p* = 0.904, respectively; veins: *p* = 0.641, *p* = 0.859, respectively). (**C**) In the group with recurrence of macular edema (recurrent group, *n* = 23), we found no significant differences in RFV between baseline and 1 month after IRI in the arteries of the non-occluded and occluded regions (*p* = 0.540, *p* = 0.090, respectively) or veins of the non-occluded region (*p* = 0.638); however, we did find a significant difference in RFV between baseline and 1 month after IRI in the veins of the occluded region (* *p* <0.001).

**Figure 3 jcm-10-00058-f003:**
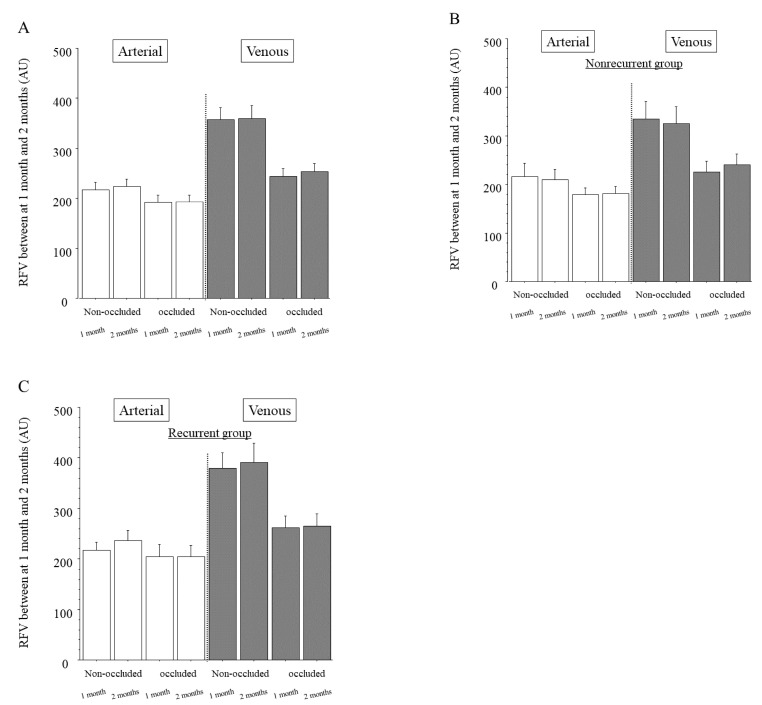
Relative blood volume between the 1 and 2 month time points after intravitreal ranibizumab injection. (**A**) In the whole sample, we found no significant differences in relative flow volume (RFV) between 1 and 2 months after intravitreal ranibizumab injection (IRI) in the arteries of the non-occluded and occluded regions (*p* = 0.575, *p* = 0.908, respectively). We also found no significant differences in RFV between 1 and 2 months after IRI in the veins of the non-occluded and occluded regions (*p* = 0.947, *p* = 0.316, respectively). (**B**) In the group with no recurrence of macular edema (nonrecurrent group, *n* = 24), we found no significant differences in RFV between 1 and 2 months after IRI in the arteries and veins of the non-occluded and occluded regions (arteries: *p* = 0.172, *p* = 0.996, respectively; veins: *p* = 0.702, *p* = 0.784, respectively). (**C**) In the group with recurrent macular edema (recurrent group, *n* = 23), we found no significant differences in RFV between 1 month (before the first recurrence) and 2 months (time of recurrence) after IRI in the arteries and veins of the non-occluded and occluded regions (arteries: *p* = 0.736, *p* = 0.877, respectively; veins: *p* = 0.777, *p* = 0.243, respectively).

**Figure 4 jcm-10-00058-f004:**
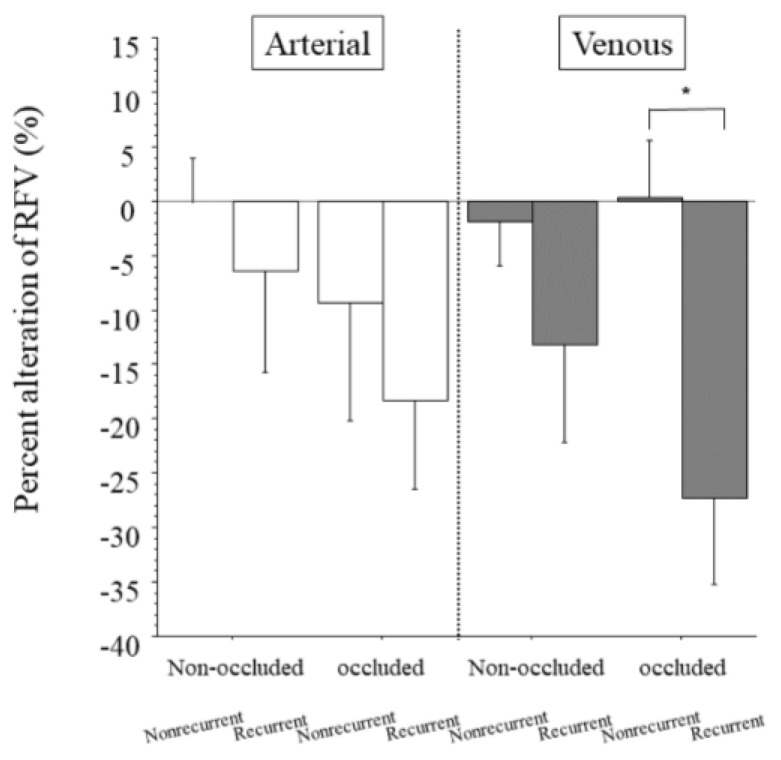
Percent change in relative blood volume between patients with and without recurrent macular edema. We found no significant differences between the nonrecurrent and recurrent groups in the percent change in RFV in the arteries of the non-occluded and occluded regions (*p* = 0.525, *p* = 0.512, respectively). We found no significant difference in the percent alteration in RFV between the nonrecurrent and recurrent groups in the veins of the non-occluded region (*p* = 0.251), but we did find a significant difference in the percent alteration in RFV between the groups in the veins of the occluded region (* *p* = 0.005).

**Table 1 jcm-10-00058-t001:** Baseline clinical features of the resolved and recurrent groups.

Findings	Resolved Group (*n* = 24)	Recurrent Group (*n* = 23)	*p* Value
Age (years)	63.4 ± 11.8 ^‡^	67.0 ± 7.2 ^‡^	0.213
Gender (female/male)	16/8	11/12	0.192
Duration of macular edema (days)	45.7 ± 29.0 ^‡^	39.1 ± 15.8 ^‡^	0.338
Hypertension	15	19	0.123
Systolic Blood pressure (mmHg)	139 ± 18	144 ± 13	0.284
Diastolic Blood pressure (mmHg)	83 ± 13	87 ± 10	0.239
Hyperlipidemia	10	12	0.471
Baseline BCVA (logMAR)	0.44 ± 0.40 ^‡^	0.46 ± 0.36 ^‡^	0.806
Baseline CMT (μm)	611 ± 211 ^‡^	666 ± 194 ^‡^	0.353
MBP (mmHg)	102 ± 13 ^‡^	106 ± 9.9 ^‡^	0.191
OPP (mmHg)	54 ± 9.1 ^‡^	57 ± 6.4 ^‡^	0.240

BCVA = best-corrected visual acuity; CMT = central macular thickness; log MAR = logarithm of the minimum angle of resolution; MBP = mean blood pressure; OPP = ocular perfusion pressure; ^‡^ Mean ± standard deviation (SD).

**Table 2 jcm-10-00058-t002:** Aqueous humor levels of factors/cytokines and the aqueous flare value at baseline.

Aqueous Factors/Cytokines	sVEGFR-1 (pg/mL)	sVEGFR-2 (pg/mL)	VEGF (pg/mL)	PlGF (pg/mL)	PDGF-AA (pg/mL)	sICAM-1 (pg/mL)	MCP-1 (pg/mL)	IL-6 (pg/mL)	IL-8 (pg/mL)	IL-12 (p70) (pg/mL)	IL-13 (pg/mL)	IP-10 (pg/mL)
At Baseline (median (interquartile range))	682 (364–1412)	491 (382–696)	63.3 (12.4–136)	3.13 (2.04–5.37)	22.1 (14.1–30.1)	3.64 (0.51–10.5)	1212 (971–1822)	3.64 (0.64–7.76)	13.8 (6.20–23.6)	0.61 (0.14–1.29)	0.48 (0.11–1.03)	73.4 (39.5–138)
Variable	*r* *p* value	*r* *p* value	*r* *p* value	*r* *p* value	*r* *p* value	*r* *p* value	*r* *p* value	*r* *p* value	*r* *p* value	*r* *p* value	*r* *p* value	*r* *p* value
Arterial of the non-occluded region	−0.01 0.983	−0.14 0.409	−0.19 0.246	0.09 0.576	−0.22 0.187	0.08 0.648	−0.14 0.416	0.03 0.849	−0.08 0.619	0.01 0.991	0.07 0.677	−0.10 0.569
Arterial of the occluded region	−0.04 0.793	0.11 0.502	−0.23 0.152	−0.09 0.615	−0.17 0.325	0.12 0.464	−0.17 0.321	−0.10 0.533	−0.14 0.391	0.02 0.887	0.09 0.566	−0.29 0.072
Venous of the non-occluded region	−0.06 0.743	−0.06 0.721	−0.17 0.303	0.19 0.248	0.10 0.551	0.16 0.342	0.07 0.684	0.11 0.503	0.14 0.412	0.13 0.423	0.23 0.158	−0.03 0.853
Venous of the occluded region	−0.04 0.818	−0.06 0.731	−0.18 0.275	−0.17 0.299	−0.14 0.405	−0.02 0.892	−0.32 0.029	−0.13 0.439	−0.31 0.035	0.08 0.616	0.01 0.941	−0.30 0.039

sVEGFR = soluble vascular endothelial growth factor receptor; VEGF = vascular endothelial growth factor; PlGF = placental growth factor; PDGF = platelet-derived growth factor; sICAM = soluble intercellular adhesion molecule; MCP = monocyte chemotactic protein; IL = interleukin; *r* = correlation coefficient. Spearman’s rank-order correlation coefficients were calculated.

**Table 3 jcm-10-00058-t003:** Aqueous humor levels of factors/cytokines and the aqueous flare value at recurrence.

Aqueous Factors/Cytokines	sVEGFR-1 (pg/mL)	sVEGFR-2 (pg/mL)	VEGF (pg/mL)	PlGF (pg/mL)	PDGF-AA (pg/mL)	sICAM-1 (pg/mL)	MCP-1 (pg/mL)	IL-6 (pg/mL)	IL-8 (pg/mL)	IL-12 (p70) (pg/mL)	IL-13 (pg/mL)	IP-10 (pg/mL)
At Recurrence (median (interquartile range))	493 (274–912)	503 (333–643)	49.5 (1.60–105)	2.64 (2.02–5.26)	20.2 (13.8–28.2)	3.64 (1.20–8.35)	1139 (909–1362)	1.75 (0.64–3.13)	11.5 (8.13–18.5)	0.61 (0.22–2.43)	0.48 (0.10–1.62)	74.2 (42.2–92.3)
Variable	*r* *p* value	*r* *p* value	*r* *p* value	*r* *p* value	*r* *p* value	*r* *p* value	*r* *p* value	*r* *p* value	*r* *p* value	*r* *p* value	*r* *p* value	*r* *p* value
Arterial of the non-occluded region	−0.04 0.873	0.30 0.231	0.02 0.943	−0.03 0.919	−0.13 0.601	0.08 0.001	0.46 0.753	−0.04 0.862	−0.03 0.906	0.11 0.656	0.07 0.788	0.01 0.961
Arterial of the occluded region	−0.04 0.865	0.29 0.178	−0.20 0.427	−0.11 0.671	−0.16 0.528	0.33 0.182	0.30 0.222	−0.03 0.918	−0.01 0.957	0.08 0.761	0.19 0.443	0.01 0.987
Venous of the non-occluded region	−0.12 0.628	0.20 0.349	−0.27 0.277	−0.04 0.871	0.10 0.691	0.20 0.440	0.24 0.341	0.20 0.436	0.11 0.662	0.21 0.404	0.19 0.381	0.18 0.461
Venous of the occluded region	−0.04 0.879	0.10 0.703	−0.03 0.921	−0.23 0.358	0.26 0.302	0.01 0.956	0.04 0.887	0.16 0.521	0.03 0.919	0.36 0.085	0.15 0.483	0.24 0.330

sVEGFR = soluble vascular endothelial growth factor receptor; VEGF = vascular endothelial growth factor; PlGF = placental growth factor; PDGF = platelet-derived growth factor; sICAM = soluble intercellular adhesion molecule; MCP = monocyte chemotactic protein; IL = interleukin; *r* = correlation coefficient. Spearman’s rank-order correlation coefficients were calculated.

## Data Availability

Data will not be shared because the authors are performing other analyses that have not yet been published.

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
