# Peer review of "Retinal Microcirculation and Cytokines as Predictors for Recurrence of Macular Edema after Intravitreal Ranibizumab Injection in Branch Retinal Vein Occlusion"

_jcm, 2020, doi:10.3390/jcm10010058_

Round 1

Reviewer 1 Report

Noma et al reported an interesting article in which they concluded that a decrease of RFV in the veins of the occluded region 35 might cause the recurrence of macular edema and that the recurrence might depend on the change 36 in RFV in the veins of the occluded region rather than the levels of cytokines. This is a potentially interesting study, but to improve the text, I suggest including more detailed information that answers the following questions: 

  • I think it would be good to have more aspects about the acceptance of the study by ethics committees. Is it a trial? 
  • I am not clear about the selection of the subjects. This is an important aspect due to it can lead to a lot of variability in the results. More information about the sample is needed
  • In the discussion it is necessary to highlight the advantages of this new strategy. What does it bring?
  • The strengths and weaknesses of this study are missing from the discussion. What is its application in clinical practice? Does it?
  •  

Author Response

Response to Reviewer 1 Comments

Noma et al reported an interesting article in which they concluded that a decrease of RFV in the veins of the occluded region might cause the recurrence of macular edema and that the recurrence might depend on the change in RFV in the veins of the occluded region rather than the levels of cytokines. This is a potentially interesting study, but to improve the text, I suggest including more detailed information that answers the following questions: 

Thank you for your review and useful comments to improve our manuscript, below we provide our point-by-point responses.

Point 1: I think it would be good to have more aspects about the acceptance of the study by ethics committees. Is it a trial? 

Response 1: As suggested, the registration number referencing this study’s approval by the ethics committees has been added to the Methods section (page 2, lines 74, 76, and 77).

Point 2: I am not clear about the selection of the subjects. This is an important aspect due to it can lead to a lot of variability in the results. More information about the sample is needed

Response 2: For the selection of subjects, all consecutive patients with BRVO who presented to Tokyo Medical University between June 2017 and July 2019 were screened using the criteria listed below. A final 47 patients with BRVO (47 eyes), who were scheduled to undergo IRI (Lucentis; 0.5 mg in 0.05 ml; Genentech, Inc., South San Francisco, CA), were included. This information has been added to the Methods section (page 2, lines 78-81, and 83-87).

Point 3: In the discussion it is necessary to highlight the advantages of this new strategy. What does it bring?

Response 3: As a new strategy, measuring RFV in the veins of the occluded region by LSFG may help predict the recurrence of macular edema. In terms of clinical practice, if RFV decreases, as measured with LSFG, taking circulatory drugs might help prevent a decrease in RFV. This information has been added to the Discussion section (page 9, lines 278-280).

Point 4: The strengths and weaknesses of this study are missing from the discussion. What is its application in clinical practice? Does it?

Response 4: As commented above, the strength of this study is that measuring RFV in the veins of the occluded region by LSFG may help predict the recurrence of macular edema. Moreover, for clinical practice, if RFV decreases, as measured with LSFG, taking circulatory drugs might help prevent a decrease in RFV. However, further investigation will be required to elucidate the relation between taking circulatory drugs, change in RFV, and recurrence of macular edema. This information has been added to the Discussion section (page 9, lines 278-283).

On the other hand, as the weaknesses of this study, our study population was relatively small. Also, we were not able to collect aqueous humor 2 months after IRI in the nonrecurrent group. Therefore, we were unable to investigate the difference in cytokines between nonrecurrent group and recurrent group. This information has been added in the Discussion section as a limitation (page 10, lines 336-340).

Reviewer 2 Report

The authors Show interesting results of a prospective study investigating the association of retinal flow volume in patients with BRCO before and after intravitreal injection of ranibizumab for macular edema. They found that RFVis reduced in patients with recurrent macular edema one month after ranibizumab. Several proinflammatory cytokines (MCP-1, IL-8, IP-10) were elevated in patients with lower RVF.

There are some issues / questions, which should be adressed:

  • The authors describe a causality between RFV and recurrent macular edema. With the data provided, I would rather suggest to describe an association than a causal dependency. The authors clearly show their valuable thoughts on a possible dependency in the discussion. However, to me it is still not convincing that the (reduced) RFV at month 1  (but not 2) would be causing a recurrent ME. Please consider.
  • Similarly, the authors propose that RFV might cause vasoconstriction (ll. 293-4) - might it not be the other way round?
  • Please elaborate on the inclusion criteria. Was pretreatment with steroids (topical, systemic) allowed or performed? Were patients with anti-inflammatory comedication excluded?
  • How and when were aqueous samples during follow-up collected? How much fluid was obtained?
  • Was there a standard, in which segment the measurements ("squares") of the RFV were done? Does the analysis depend on a correct positioning, or do different segments of the same vessel show different values?

Author Response

Response to Reviewer 2 Comments

The authors Show interesting results of a prospective study investigating the association of retinal flow volume in patients with BRCO before and after intravitreal injection of ranibizumab for macular edema. They found that RFVis reduced in patients with recurrent macular edema one month after ranibizumab. Several proinflammatory cytokines (MCP-1, IL-8, IP-10) were elevated in patients with lower RVF. 

There are some issues / questions, which should be adressed:

Thank you for reviewing our manuscript and the useful comments to improve our manuscript. Below we provide our point-by-point responses.

Point 1: The authors describe a causality between RFV and recurrent macular edema. With the data provided, I would rather suggest to describe an association than a causal dependency. The authors clearly show their valuable thoughts on a possible dependency in the discussion. However, to me it is still not convincing that the (reduced) RFV at month 1  (but not 2) would be causing a recurrent ME. Please consider.

Response 1: As suggested, a causality between RFV and recurrent macular edema was overstated. Therefore, the wording regarding causality has been revised throughout (page 1, line 36; page 8, line 268; page 9, line 276; page 10, line 347).

Point 2: Similarly, the authors propose that RFV might cause vasoconstriction (ll. 293-4) - might it not be the other way round?

Response 2: As suggested, this expression was also overstated.  Therefore, the expression has been revised (page 9, line 312).

Point 3: Please elaborate on the inclusion criteria. Was pretreatment with steroids (topical, systemic) allowed or performed? Were patients with anti-inflammatory comedication excluded?

Response 3: In this study, pretreatment with steroids (topical, systemic) was not performed. Therefore, patients with anti-inflammatory comedication were excluded. This information has been added to the Methods section (page 2, lines 86 and 87).

Point 4: How and when were aqueous samples during follow-up collected? How much fluid was obtained?

Response 4: Information regarding collection of aqueous samples has been added to the Methods section (page 3, lines 95-97).

Point 5: Was there a standard, in which segment the measurements ("squares") of the RFV were done? Does the analysis depend on a correct positioning, or do different segments of the same vessel show different values?

Response 5: In this study, we confirmed that the positioning for measurements at 1 and 2 months was the same positioning as at baseline for all patients. Furthermore, it has been reported that RFV measurements have high reproducibility regarding measurement error. This information has been added to the Methods section (page 4, lines 130-133).